# Model Re-Estimation: An Alternative for Poor Predictive Performance during External Evaluations? Example of Gentamicin in Critically Ill Patients

**DOI:** 10.3390/pharmaceutics14071426

**Published:** 2022-07-07

**Authors:** Alexandre Duong, Chantale Simard, David Williamson, Amélie Marsot

**Affiliations:** 1Faculté de Pharmacie, Université de Montréal, Montreal, QC H3T 1J4, Canada; david.williamson@umontreal.ca (D.W.); amelie.marsot@umontreal.ca (A.M.); 2Laboratoire de Suivi Thérapeutique Pharmacologique et Pharmacocinétique, Faculté de Pharmacie, Université de Montréal, Montreal, QC H3T 1J4, Canada; 3Institut Universitaire de Cardiologie et Pneumologie de Québec, Quebec, QC G1V 4G5, Canada; chantale.simard@pha.ulaval.ca; 4Faculté de Pharmacie, Université Laval, Quebec, QC G1V 0A6, Canada; 5Hôpital Sacré-Cœur de Montréal, Université de Montréal, Montreal, QC H4J 1C5, Canada; 6Centre de Recherche, CHU Sainte Justine, Montreal, QC H3T 1C5, Canada

**Keywords:** gentamicin, population pharmacokinetic modeling, external evaluation, model re-estimation, dosing nomogram

## Abstract

Background: An external evaluation is crucial before clinical applications; however, only a few gentamicin population pharmacokinetic (PopPK) models for critically ill patients included it in the model development. In this study, we aimed to evaluate gentamicin PopPK models developed for critically ill patients. Methods: The evaluated models were selected following a literature review on aminoglycoside PopPK models for critically ill patients. The data of patients were retrospectively collected from two Quebec hospitals, the external evaluation and model re-estimation were performed with NONMEM^®^ (v7.5) and the population bias and imprecisions were estimated. Dosing regimens were simulated using the best performing model. Results: From the datasets of 39 and 48 patients from the two Quebec hospitals, none of the evaluated models presented acceptable values for bias and imprecision. Following model re-estimations, all models showed an acceptable predictive performance. An a priori dosing nomogram was developed with the best performing re-estimated model and was consistent based on recommended dosing regimens. Conclusion: Due to the poor predictive performance during the external evaluations, the latter must be prioritized during model development. Model re-estimation may be an alternative to developing a new model, especially when most known models display similar covariates.

## 1. Introduction

Gentamicin is a broad-spectrum antibiotic from the aminoglycoside family mostly used against life-threatening infections due to suspected Gram-negative bacteria [1,2]. The antimicrobial activity of gentamicin, along with other aminoglycosides, is concentration-dependent; therefore, its efficacy is based on the peak serum level (C_max_) or the area under the concentration curve (AUC) related to the minimal inhibitory concentration (MIC) [3]. Moreover, due to the known potential ototoxicity and nephrotoxicity caused by aminoglycoside administration, therapeutic drug monitoring (TDM) is essential to achieve pharmacokinetic/pharmacodynamic (PK/PD) targets whilst minimizing toxicity. Considering the narrow therapeutic index of aminoglycosides, the administration of aminoglycosides has slowly shifted from a multiple daily dose (MDD) to a once-daily dose (ODD) throughout the years. The latter, also known as extended-interval dosing, has shown better signs of minimizing toxicity whilst also maintaining efficacy endpoints [4,5]. These PK/PD endpoints may be more difficult to attain in several frail populations such as critically ill patients. Due to their severe pathophysiological changes, standard dosing regimens may lead to inadequate concentrations and clinical outcomes. Therefore, the implementation of TDM based on population pharmacokinetic (PopPK) models in a clinical routine for critically ill patients should be prioritized, especially considering their high mortality rates [6].

In order to better understand aminoglycoside pharmacokinetics and the optimization of drug administration in critically ill patients, multiple PopPK models for gentamicin have been developed throughout the years [7]. Most of them did not include an external evaluation during the model development. An external evaluation, one of the most robust validation methods and a key step before the clinical application, consists of using an independent population within the final model to assess the accuracy and reproducibility of predicting the antimicrobial concentrations and clinical outcomes [8].

The primary objectives of this study were to evaluate previously published gentamicin PopPK models within a population of critically ill patients and to determine their predictive performances in order to use them during TDM in clinical settings. The subsequent objective was to determine the best performing model dosing regimens with simulations.

## 2. Materials and Methods

### 2.1. Patients

The medical records of adult ICU patients admitted to the Hôpital Sacré-Cœur de Montréal (HSCM) between 2009 and 2019 or the Institut universitaire de cardiologie et pneumologie de Québec (IUCPQ) between 2014 and 2020 and who received at least 1 dose of gentamicin and 1 serum concentration were retrospectively reviewed. Multicenter ethics approval was obtained from the Comité d’Éthique du CIUSSS-du-Nord-de-l’Île-de-Montréal (CERC-19-073-R (1) and HSCM: MP-32-2020-1904).

Data extraction from the medical records included age, sex, serum creatinine, body weight, gentamicin dose administered, gentamicin serum concentrations and infusion time dates as well as the times of all doses and concentrations, concomitant medications, medical history and admission diagnoses. Creatinine clearance based on Cockcroft–Gault (CLCG) and the glomerular filtration rate (eGFR) were estimated based on the closest time of the serum creatinine measurement according to the respective equations [9,10]:eGFR (mL/min) = 186.3 × (Scr/88.4)^(−1.154)^ × Age^(−0.203)^ × (1.212 if black) × (0.742 if female)(1)
CrCl (mL/min) = ((140 − Age) × Body weight (kg) × 1.23 × (0.85 if female))/Scr(2)

### 2.2. Published Models

A literature review of aminoglycoside PopPK models for critically ill patients was previously performed. In this current study, we only aimed to externally evaluate the gentamicin PopPK models; therefore, all gentamicin PopPK models that were developed using non-linear mixed effect modeling (NONMEM) software were included in the external evaluation. Models were excluded if information on the pharmacokinetic equations of the models was missing in the respective article.

### 2.3. Model Evaluation

The external evaluation was conducted using NONMEM^®^ (version 7.5: ICON Development Solutions, Ellicott City, MD, USA) and the plots were designed using R version 4.0.4. The evaluation was performed by combining both datasets (HSCM and IUCPQ).

The retained PopPK models were described based on the formulas and PK parameters reported from the final model for each publication. If a required covariate was not available within the datasets, it was assigned with the typical value of the model. No additional fitting was used during the external evaluation (the option in NONMEM was set to MAXEVAL = 0). The global fit of the PopPK models was also assessed with goodness-of-fit (GOF) plots of the predicted concentrations versus the observed concentrations. The predictive performance of the models was evaluated with the prediction error (PE) determined by the following equation:PE (%) = (C_(pred,i)_ − C_(obs,i)_)/C_(obs,i)_ × 100%(3)
where C_pred_ and C_obs_ correspond with the *i*th predicted concentration by the model and the observed concentration, respectively [11]. To quantify the bias and inaccuracy, the median prediction error (MDPE) and median absolute prediction error (MADPE) were used with the following equations:Bias: MDPE_i_ (%) = median (PE_ij_, j = 1, …, Ni)Inacurracy: MADPE_i_ (%) = median (|PE_ij_|, j = 1, …, N_i_)

In order to be considered unbiased, the MDPE should be between −20 and 20% whereas to be considered accurate, the MADPE value should be ≤ 30% [12]. Finally, we used a normalized prediction distribution error (NPDE) analysis as a strategy to establish the overall fit of the PopPK model with the independent databases.

### 2.4. Model Re-Estimation

In the instance of an inadequate predictive performance of the models following an external evaluation based on the abovementioned criteria, the PK parameters and interindividual variability were re-estimated by NONMEM using the combined datasets of HSCM and IUCPQ. The re-estimated parameters were compared with the original values and the overall fit of the GOF plots. Normalized prediction distribution errors (NPDEs) as well as the corresponding statistical tests for normal distribution and homogeneity of variance and bootstraps were also assessed. If the re-estimated models were successfully minimized, the population and individual bias and imprecision were calculated and compared.

### 2.5. Simulations of C_max_/MIC > 8–10 and C_min_ < 1 or 0.5 mg/L Following a Third Dose

Considering that the clinical efficacy of gentamicin as well as other aminoglycosides is based on C_max_/MIC, the prediction of peak concentration following the third dose was assessed (C_max,3rd_) using different dosing regimens for the best performing PopPK model. The pre-dose concentration before the fourth was also examined. These simulated PK/PD endpoints were obtained based on the covariates of the patients only (a priori prediction). The evaluation of these simulated concentrations was only completed with the best performing PopPK model in terms of the overall predictive performance (GOF plots, MDPE and MADPE).

## 3. Results

The medical records of 48 and 39 ICU patients from IUCPQ and HSCM, respectively, were retrieved for this study. Table 1 describes the demographic characteristics of both populations, separately and altogether. The only demographic characteristics that were statistically different between both institutions were sex and serum creatinine (Scr). Moreover, the total daily dose at HSCM appeared to be higher and more variable than at IUCPQ. In fact, IUCPQ mostly generalized their care toward people with cardiopulmonary diseases; therefore, the majority of the patients from IUCPQ included in this study suffered from endocarditis whereas the patients from HSCM suffered a variety of conditions mostly leading to sepsis.

From our literature review of gentamicin PopPK models, eleven models were screened for inclusion [7]. Amongst them, seven were excluded due to a lack of information or if the models were not developed with NONMEM (n = 7). Therefore, the predictive performance of four models was evaluated [13,14,15,16]. The demographic characteristics are presented in Table 1. The pharmacokinetic equations of the four evaluated models are presented in Appendix A. Half of them used mono-compartment models whereas the other half used bi-compartment models [13,14,15,16]. The covariates used were varied, with the use of glomerular filtration in one study, CrCl in another study, the covariates related to weight in two studies and albumin in one study. Gentamicin CL and the total volume of distribution ranged between 1.15 and 5.7 L/h and 19 and 54 L, respectively (Appendix A).

The results presented in this article were from an external evaluation using combined datasets from HSCM and IUCPQ. The external evaluations were performed separately for each institution and obtained comparable results. The population-predicted versus the observed concentrations are presented in Appendix A for each evaluated model. The models from Bos et al. and Hodiamont et al. appeared to underpredict the observed concentrations [14,16] whereas the models from Rea et al. and Hodiamont et al. tended to overpredict the observed concentrations from the validated dataset [13,15]. Following the external evaluation, the population bias and imprecision values ranged between −44.0 and 66.1% and 47.8 and 69.9%, respectively. The bias and imprecision values improved when individual characteristics were taken into consideration, with values ranging between −18.0 and 10.1% and 18.0 and 27.1%, respectively.

As presented in Table 2, all four models evaluated did not respect the targeted ranges for population bias (±20%) and imprecision (≤30%) [12]. These models were, therefore, re-estimated in NONMEM with the combined datasets of HSCM and IUCPQ. The PK parameter coefficients in their respective PK equations are presented in Appendix A for the external evaluation and the re-estimation, respectively. The typical PK parameters considered during the external evaluation and the re-estimated typical PK parameters as well as their respective interindividual variability are presented in Appendix A.

Upon the model re-estimation, the difference between the new PK parameter values and the respective original values was generally greater for the mono-compartmental models [13,14] than for the bi-compartmental models [15,16]. For the model of Rea et al., the typical re-estimated gentamicin clearance was around 50% greater than its original value used in the external evaluation whereas the re-estimated volume of distribution was half its original value [13]. For the model of Bos et al., the typical re-estimated gentamicin clearance was slightly lower than its original value whereas the re-estimated volume of distribution was slightly higher than its original value. For the first model of Hodiamont et al. [15], both the gentamicin clearance and total volume of distribution were higher following the re-estimation compared with the original values. For the second model of Hodiamont et al. [16], both the gentamicin clearance and total volume of distribution were lower following the re-estimation compared with the original values.

The interindividual variability appeared to be generally lower with the re-estimated PopPK models. All models were successfully minimized; the population and individual bias and imprecision from these re-estimated models are presented in Table 2. The population-predicted versus the observed concentrations are presented in Figure 1 for each re-estimated model. Following the model re-estimation, the population-predicted concentrations drastically improved for each model compared with its own counterpart during the external evaluation.

Only the re-estimated model of Rea et al. was able to adequately predict the observed concentrations with an acceptable population bias and imprecision. Although the re-estimated model of Bos et al. had a population imprecision of 30.1%, its individual imprecision increased following the re-estimation from 18.0% to 20.4%. Therefore, the re-estimated model from Rea et al. was deemed to be the best performing model and was used for the therapeutic target simulations. The normalized prediction distribution errors (NPDEs) were also compared between the original and re-estimated PK parameters, as shown in Appendix A. For the original model, the statistical test results showed a normal distribution (*t*-test of 0.507), but a heterogeneity of variance (Fisher of <0.001). Although the statistical tests showed a non-normal distribution (*t*-test of 0.0371) and heterogeneity of variance (Fisher of 0.0495), the graphical representations of the NPDE (Q-Q plot and histogram) showed a better distribution and the bootstraps results were adequate (Appendix A).

Both therapeutic targets (C_max,3rddose_ and pre-dose before the fourth administration) were simulated for several dosing regimens (MDD and ODD). The simulations were based on two different efficacy targets: C_max_/MIC > 8 (Appendix A) and C_max_/MIC > 10 (Appendix A). Figure 2 presents the probability of target attainment (PTA) based on the MIC values and the dosing regimen used. Appendix A presents the same PTA, but displayed by total dose given per day for C_max_/MIC > 8. Appendix A displays the percentage for the pre-dose concentrations before the fourth administration below 1 mg/L or 0.5 mg/L. Similarly, Appendix A presents the same percentage, but displayed by dosing regimen.

For each of the eight simulated doses given per day (3 to 12 mg/kg/day), the once-daily dosing regimen was the best dosage in order to maximize the probability of target attainment for all MIC values compared with the multiple daily dosing regimen (twice or thrice daily). Similarly, for the pre-dose concentrations before the fourth administration, a higher dosing interval led to a higher probability of respecting the toxicity targets.

For severe infections, as per the latest MIC breakpoints from the United States Committee on Antimicrobial Susceptibility Testing (USCAST) and the European Committee on Antimicrobial Susceptibility Testing (EUCAST), *Staphylococcus* spp., *Pseudomonas* spp. and *Enterococcus* spp. present MIC values ranging from 1 to 2 mg/L. Considering these actual MIC values, only dosing regimens greater than 5 mg/kg/day had a greater PTA of 90% for an MIC value of 1 mg/L. Although these dosing regimens should reach efficacy targets of C_max_/MIC > 8, toxicity targets should be cautiously monitored. Around 50% and 64.4% of the patients presented C_trough_ before the fourth administration greater than 1 mg/L with a dosing regimen of 5 mg/kg/day and 12 mg/kg/day, respectively (Appendix A). Simulations of C_trough_ before the fourth administration for multiple daily dosing regimens (twice and thrice daily) led to poor percentages of target attainment.

## 4. Discussions

In the past decades, multiple PopPK models of gentamicin for critically ill patients have been developed [7]. In this current study, we evaluated the predictive performance of four models using an independent dataset with medical records from two hospitals [13,14,15,16]. The model appropriateness was evaluated based on an integrative assessment of several markers such as bias, imprecision and GOF plots. Based on the population bias and imprecision values, all four models were not within the predefined values, thereby suggesting that all four models are not directly transferable to a clinical application. Moreover, based on the observed versus the predicted concentrations from the models, the four models showed greater under- or overpredictions of the observed gentamicin concentrations. The underprediction and overprediction of actual therapeutic drug monitoring concentrations can result in a misinterpretation of efficacy and toxicity targets, respectively. This poor population prediction may have been due to the differences in the demographic characteristics and clinical conditions from the respective population of the models and the Quebec population. However, the individual prediction performance, as shown in Table 2, improved enough to be within an acceptable range, suggesting that the use of all four models may be feasible in a posteriori dosing adaptation.

If the evaluated models showed a poor predictive performance with our population, two options were considered: to develop a PopPK model with our database or to simply adjust the pre-existing models. Considering that the evaluated models included covariates that were available within our population, we opted for a re-estimation of these models. The latter method was expected to improve the predictive performance due to the adaptation of the PK parameters of each model based on our population.

Due to the differences in our population compared with the respective populations of each model, the PK parameter estimates varied following the model re-estimation. For instance, a typical clearance value of the re-estimated model from Rea et al. was around 50% higher than its respective original value. This may be explained by the eGFR value of our combined populations (HSCM + IUCPQ) being around 60% higher than the population used to develop the model from Rea et al. [13]. Moreover, this covariate was also deemed to be significant to determine gentamicin clearance. In parallel, although the body weight was similar between Rea et al. and our study population, the re-estimated volume of distribution was half the value of the original model. Considering that critically ill patients often suffer from fluid overload caused by complications [17], this may explain the higher volume of distribution compared with the endocarditis patients from IUCPQ in our combined datasets. For the model of Bos et al., the re-estimated gentamicin clearance was lower than its original value, leading to an underprediction of the gentamicin concentration during the external evaluation. This may be due to the differences in the severity of the medical conditions and demographic characteristics between both populations. Although the population used to develop the model of Bos et al. was severely ill, it was noted that they were not in ICU whereas our patients were hospitalized in ICU settings. Moreover, the age and body weight from the sub-Saharan African population, which were both considered in the Cockcroft–Gault calculation of CrCl, were significantly lower than our population. As for the model of Hodiamont et al. [16] although the body weight was not statistically different between their respective population and our two populations, the typical volume of distribution of the re-estimated model was half than its original value. Only 7% of their population had endocarditis compared with 55% in our dataset. A higher proportion of critically ill patients where fluid overload often occurs due to sepsis may suggest the higher distribution volume observed in the study population of Hodiamont et al. [16].

Although the re-estimation of PopPK models with our population improved the predictive performance of all models, a variability remained in the prediction of the actual gentamicin concentrations as well as in the re-estimated PK parameters. Our external validation datasets formed from our two institutions consisted of patients with severe infections or endocarditis, which was comparable with the populations used to develop the evaluated models [13,14,15,16]. This variability could have been caused by several sources such as the severity of the illness, medical history and related concomitant medication. Furthermore, the origin of the study populations of the developed models was varied, with patients from the United States, Africa or Europe. The variability may also have been due to the differences between the patients of the developed PopPK model. As shown in Appendix A, the interindividual and residual variabilities for the PK parameters were already high, thereby suggesting that the patients from the original dataset used in the model development were different from each other.

The variability also seen during the external evaluation may also have been due to the different study designs from the developed PopPK models. The number of patients in the validation dataset was greater than most study populations used for the model development [14,15,16]. Moreover, three studies had a similar sampling schedule to the validation datasets with samples from therapeutic drug monitoring [13,15,16] whereas the model from Bos et al. [14] was developed with samples collected following a prospective observational design. Alihodzic et al. demonstrated that erroneous records due to clinical routine practices may lead to an inaccurate estimation of PK parameters during PopPK model development [18].

Based on the best re-estimated performing model, we developed an a priori dosing nomogram based on the different MIC values, dosing regimens and dosing intervals. For the evaluations of C_max_ and C_min_ following the third gentamicin administration, the ODD regimen appeared to be the best option in order to maximizes the PTA of the efficacy (C_max_/MIC > 8) and toxicity (C_min_ < 1 mg/L or C_min_ < 0.5 mg/L) targets compared with the MDD regimen. This finding was consistent with previous literature that stated that ODD regimens were able to maintain efficacy whilst minimizing the signs of toxicity [4,5].

For Gram-positive infections, the peak concentrations should be targeted at around 3 to 4 mg/L [19,20]. The latter was represented in our dosing nomogram with C_max_/MIC > 8 considering an MIC of 0.5 mg/L (Appendix A). Taken daily, a PTA over 90% is possible with any doses greater than 3 mg/kg. In terms of toxicity, the latter also represents the dosing regimen recommended for Gram-positive infections [19,20]. Although the original PopPK model of Rea et al. was not deemed to be adequate following the external evaluation, our dosing regimens simulated from its re-estimated model were in line with the literature. This finding brings to light the relevance and accuracy of the metrics generally used during an external evaluation of PopPK models.

Rea et al. also performed dosing regimen simulations with PTA based on their final model [13]. From their simulations, the probability of attaining C_max_/MIC > 10 considering an MIC value of 0.5 mg/L with a daily dose of 7 mg/kg was 87.9%. Based on our dosing nomogram, a daily dose of 7 mg/kg with an MIC value of 0.5 mg/L led to 98.2% of our patients attaining the target of C_max_/MIC > 10 (Appendix A). Consequently, the original dosing regimens recommended by Rea et al. would be higher than needed for our population.

Several limitations should be considered in this current study. Firstly, the concentrations from the medical records from both institutions were therapeutic drug monitoring data collected during a clinical setting. Therefore, the number of samples per patient was limited. Considering the retrospective design of this study, the severity of the conditions of the patients was unobtainable from the medical records as well as other covariates of interest. Moreover, choosing NONMEM software as an inclusion criterion may have restricted the number of models to be evaluated.

## 5. Conclusions

In this study, we aimed to evaluate gentamicin PopPK models with two Quebec critically ill populations. Although the four evaluated models showed a poor population predictive performance, their respective predictive performances when considering the characteristics and dosing information of the patients were adequate. In the scenario of a poor predictive performance, a model re-estimation is a viable option in order to avoid the development of PopPK models similar to pre-existing ones. With the best performing re-estimated model from Rea et al., the dosing regimens were simulated with our study population. These findings suggested that the re-estimation of existing models in order to develop an a priori dosing nomogram should be considered more often and may be more suited to each population or also used for a Bayesian analysis and estimation.

## Figures and Tables

**Figure 1 pharmaceutics-14-01426-f001:**
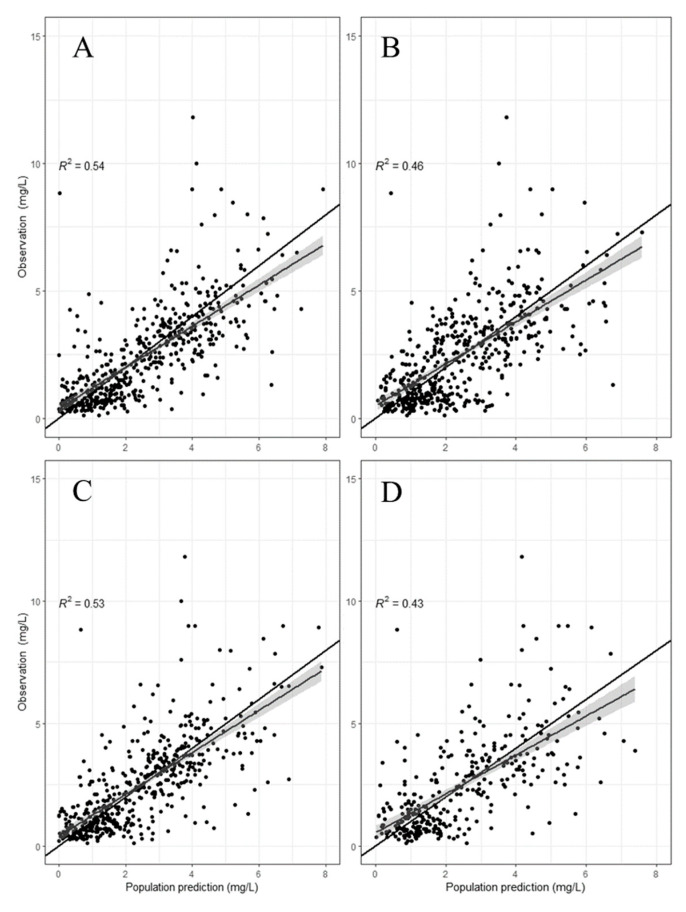
Population-predicted concentration versus observed concentrations for gentamicin models following re-estimation. (**A**) Bos et al. [14], (**B**) Hodiamont et al. [15], (**C**) Rea et al. [13], (**D**) Hodiamont et al. [16]. Black line with shaded area represents the trendline from the scatter points.

**Figure 2 pharmaceutics-14-01426-f002:**
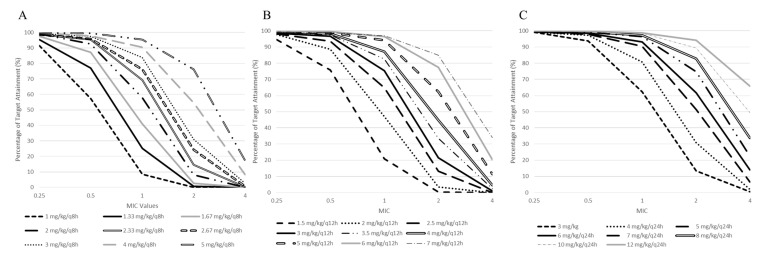
Probability of target attainment of C_max_/MIC > 8 on the third dose based on different MIC values. (**A**) Dose administered thrice daily (every 8 h). (**B**) Dose administered twice daily (every 12 h). (**C**) Dose administered daily (every 24 h).

**Table 1 pharmaceutics-14-01426-t001:** Demographic characteristics of the patients in the evaluated models and the external validation datasets.

Characteristics	Rea et al. [13]	Bos et al. [14]	Hodiamont et al. [15]	Hodiamont et al. [16]	HSCM	IUCPQ	Combined
Population type	Critically ill patients	Critically ill non-ICU sub-Saharan African adult patients	Critically ill patients on or off CVVH	Critically ill patients	Critically ill patients	Mostly endocarditis patients in ICU	Critically ill and endocarditis patients
Number of patients (N)	102	48	44	59	39	48	87
M/F	45/57	24/24	20/24	29/30	18/21	36/12	54/33
Age (years)	61.4 ± 16.4	40.0 (20–86)	61.0 (20–78)	60.9 ± 17.2	60.3 ± 19.2	58.7 ± 16.9	59.4 ± 17.9
Weight (kg)	81.4 ± 30.3	51.0 (33–76)	70.5 (42.0–116)	79.2 ± 22.0	79.4 ± 20.5	80.5 ± 22.4	80.0 ± 21.5
Serum creatinine (μmol/L)	194.5 ± 168	76.0 (37–1192)	115.0 (36–1719)	-	93.2 ± 91.4	99.9 ± 34.8	96.9 ± 66.0
CrCl (mL/min)	-	74.0 (4–155)	54.9 (4.0–150)	-	99.8 ± 60.6	86.0 ± 36.4	92.2 ± 48.9
eGFR (mL/min)	48.1 ± 26.5	-	-	-	73.5 ± 21.0	90.1 ± 39.9	80.9 ± 31.9
Albumin (g/L)	-	29 (13–40)	21.5 (10–36)	-	-	29.0 ± 5.6	29.0 ± 5.6
Total daily dose (mg/kg)	-	-	-	-	2.9 ± 0.9	2.0 ± 0.7	2.4 ± 1.1

The values are presented as median (range) or mean ± SD. CrCl: creatinine clearance; CVVH: continuous venovenous hemofiltration; eGFR: estimated glomerular filtration rate; HSCM: Hôpital du Sacré-Cœur de Montréal; IUCPQ: Institut universitaire de cardiologie et pneumologie de Québec; M/F: male/female; N: number. -: Not available

**Table 2 pharmaceutics-14-01426-t002:** Prediction error following external evaluation of the PopPK models.

Model	Population		Individual		Population (Re-Estimation)		Individual (Re-Estimation)	
	MDPE (%)	MADPE (%)	MDPE (%)	MADPE (%)	MDPE (%)	MADPE (%)	MDPE (%)	MADPE (%)
Rea et al. [13]	44.2	54.1	−18.0	27.1	2.14	28.1	−5.19	19.0
Bos et al. [14]	−44.0	47.8	−3.29	18.0	2.00	30.1	0.09	20.4
Hodiamont et al. [15]	66.1	69.9	10.1	24.0	2.20	36.9	−4.01	21.3
Hodiamont et al. [16]	−31.7	48.8	−14.7	26.8	6.03	39.2	−6.42	18.6

MDPE: median prediction error; MADPE: median absolute prediction error.

## Data Availability

Not applicable.

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
