# Peer review of "Model Re-Estimation: An Alternative for Poor Predictive Performance during External Evaluations? Example of Gentamicin in Critically Ill Patients"

_pharmaceutics, 2022, doi:10.3390/pharmaceutics14071426_

Round 1

Reviewer 1 Report

The authors have studied gentamicin pharmacokinetic (PopPK) models in critically ill patients as external evaluation which is crucial before clinical application.

to reply to this question they have studied the predictive performance of 4 models evaluated previously with an independent dataset with medical records from two hospitals in Montreal. highly data science and paper base article.

Based on their work, all the other modeling done before this article is completely acceptable.

Major comment:

Based on the principal finding of this paper, I propose to change the format of article into a “short communication”. This change allow authors explain briefly their study population and methods then they transfer their essential message in a nut shell and take homme message clearly. They can publish their modeling in a supplementary data.

Author Response

We would like to thank Reviewer #1 for their comments and time towards improving our manuscript.

Considering that this manuscript was submitted to appear in the Special Issue: “Therapeutic Drug Monitoring and Pharmacokinetics-Based Individualization of Drug Therapy”, we believe that this paper may be more suited as an original article. This format allows the readers to properly understand our study’s conceptualization and methodology while having access to our key results throughout the manuscript. However, we agree that by transferring some figures and tables from the manuscript to the Supplementary Information section (Figure 1, Table 3 and Figure 4 as per the original submitted manuscript), it may help the readers to understand the take home message with the remaining key results (Tables 1 and 2, Figures 2 and 3 as per the original submitted manuscript).

Reviewer 2 Report

I suppose that the author did not upload the supplementary materials even though it mentioned in main text. It should be better to proceed peer-review step after upload of supplementary materials first. 

Author Response

Dear Reviewer 2,

Thank you taking the time to assess our manuscript. I believe the supplementary material was already available as the other reviewer did not have problem accessing it. 

Please see attached the supplementary files.

Thank you,

Round 2

Reviewer 2 Report

The purpose of this study is to evaluate previously published gentamicin population pharmacokinetic (PopPK) models developed in critically ill patients and determine the best performing model dosing regimens with simulation. Therefore, the important thing in this paper is to evaluate models, then re-estimate models and use the best performing model to simulate different dosing regimens. The authors had done a lot of work, this reviewer recommended to go through the publication process after below minor corrections as attached file. 

Author Response

We would like to thank Reviewer #2 for their comments and time towards improving our manuscript. Please see below responses to each comment as well as corrected manuscript.

Reviewer #2comment #1:

Following the study of Teixeira-da-Silva P. et al. (2020), the external evaluation of the model using a new real-world dataset (not simulated) from a population with similar characteristics as those used to develop/build model. However, in this manuscript, parameters related to renal function were different between the demographic characteristics of the patients in four evaluated models and the external validation datasets. As I highlighted in the below table, yellow cells were matched with a normal range of renal parameters. It’s easy to see that the demographic characteristics of the patients in the four evaluated models show more severe pathophysiological changes than the external validation datasets. I think because of this reason, model evaluation would be not correct.

Response to comment #1

Thank you for your comment. We agree that the few different characteristics between model populations and our study population may have led to a poor predictive performance during external evaluations, as mentioned in the Discussion section. However, all of the models evaluated were for inpatients in intensive care units, which is why we decided to evaluate them. In addition, it should be noted that these models all had significant interindividual variability and the covariates usually identified for this molecule and population (Table S1). Given these considerations, we deemed it was still justified to perform external evaluations in the event of adequate fitting of the Pop-PK model with our study population.

Reviewer #2comment #2:

Figure 1 shows that the observed concentration in models from Bos et al [14] and Hodiamont et al [16] studies were underpredicted while in models from Rea et al [13] and Hodiamont et al [15] studies were overpredicted. However, from line 156 to 159, the content expressed the opposite trend. So, please check it again.

Response to comment #2:

Thank you for your comment, the sentence from line 156 to 159 was corrected. Also, please note that this figure was added in the Supplementary Information as per comments from Reviewer #1.

Reviewer #2comment #3:

Dosing intervals used to simulate Cmax after the 3rd dose were multiple daily dose (MDD) and one daily dose (ODD). While, in the simulation of Ctrough, dosing intervals were every 24 hours, every 36 hours and every 48 hours. And then in the discussion section, the PTA of efficacy and toxicity were compared between ODD and MDD. So, the authors need to clarify the reason for choosing different dosing intervals in the simulation of Cmax and Ctrough and what can be suggested about the dosing interval based on the result of this simulation.

Response to comment #3:

Thank you for your comment. Initially, Ctrough simulations were also performed for the dosing intervals of every 8 hours and every 12 hours (only for the dosing regimens of 3mg/kg to 7mg/kg), but they were not added due to the limited space in the manuscript.

Following your comment, we added the percentage of target attainment in Table S6 of the Supplementary Information. These poor percentage of target attainment for the dosing intervals of every 8 hours and every 12 hours further support our claim that once-daily-dose is the better dosing regimen. Considering their low percentages, they were not added in Figure S3.

Reviewer #2comment #4:

In the simulation, therapeutic targets (Cmax/MIC > 8 – 10 and Cmin < 1 or 0.5 mg/ml), timepoint (Cmax following 3rd dose, Cmin following predose before the 4th administration), dosing regimens were chosen. It would be better to provide references for these things.

Response to comment #4:

Thank you for your comment. These therapeutic targets were determined based on articles already included in the list of references. The numbers were added in the methodology section.

Reviewer #2comment #5:

When evaluated models show poor predictive performance, two options may be considered (develop a new model and re-estimate the pre-existing models). In conclusion section suggests that model re-estimation may be an alternative to developing a new model.

However, the re-estimated model used the same covariates with corresponding model and did not consider to other covariates. While, when developing a new model, covariates were evaluated and may be different with re-estimated model. So, the results may be different between two options and to confirm which one is better, the comparison need to be evaluated. The authors need to clarify the reasons and provide references for choosing model re-estimation over building a new model.

Response to comment #5:

Thank you for your comment. In our previous literature review (Duong et al.), it was shown that included significant covariates in gentamicin and aminoglycosides Pop-PK models are generally the same across the literature. Therefore, instead of adding an umpteenth Pop-PK model with similar covariates in the literature, we preferred adapting pre-existing Pop-PK models to our study population.We agree that the development of a new Pop-PK population may have been better, but dosing regimen recommendation following simulations may have remained the same.

Reviewer #2comment #6:

Reference number 19 (Nord-de-l'Île-de-Montréal Cd. Guides pratiques-Pharmacocinétique-Aminosides 2021) cannot be accessed via the attached link in the References section.

Response to comment #6:

Thank you for your comment. Following verification with multiple users, this link should be accessible.

Reviewer #2comment #7:

External evaluation and model re-estimation were performed using combining datasets in two institutions (HSCM and IUCPQ) but sex and serum creatinine of two institutions were statistically different. So, the authors need to explain the effect of this combination.

Response to comment #7:

Thank you for your comment. This abstract initially performed external evaluations with each institution separately. Despite the differences in sex and serum creatinine between HSCM and IUCPQ, external evaluations for all models were comparable between the two institutions. Therefore, we proceeded to combine both datasets into one to increase the number of patients. A sentence was therefore added in the manuscript to clarify this decision of combining both datasets. Due to the several figures and tables already present in the Supplementary Information, we think the results of external evaluation with each institution may not be necessary to be included.

Reviewer #2comment #8:

A re-estimated model that predicted adequately the observed concentrations with acceptable population bias and imprecision was chosen to simulate the dosing regimens. However, in case two or more models achieve that, what will be the criteria to choose the best model. Authors need to clarify the criteria.

Response to comment #8:

Although all models improved following model re-estimation, only the model from Rea et al. obtained results within the targeted ranges for population bias (±20%) and imprecision (≤ 30%). For the model from Bos et al., although its population imprecision was close to the targeted ranges (30.1%), the individual imprecision increased following re-estimation (from 18.0% to 20.4%), as opposed to the improvement seen in Rea’s model for individual imprecision following re-estimation (27.1% to 19.0%). A sentence was added in the results section of the manuscript to clarify this statement.

Reviewer #2comment #9:

Line 146: I think seven models were excluded from this study due to both lack of information and not development by using the nonlinear mixed-effect modeling (NONMEM) software. Here, you only mention the reason is just a lack of information.

Response to comment #9:

Thank you for your comment. It was corrected to add both reasons.

Reviewer #2comment #10:

Line 152 – 153: It’s better to provide one more table to show the exact value of gentamicin CL and the total volume of distribution of evaluated models in four studies.

Response to comment #10:

Thank you for your comment. The values of gentamicin CL and volume of distribution for the four studies were included in Table S1 of the Supplementary Information. Details were added following the line to refer the reader to Table S1.

Reviewer #2comment #11:

Line 160 – 163: I don’t see values for -56.75 to 68.29% and 48.82 to 71.88% as well as -18.02 to 10.56% and 18.86 to 28.06% in table 2 although this table show population and individual bias and inaccuracy. Please note to the table/figure mentions these values.

Response to comment #11:

Thank you for your comment. Upon revision, the values in the Table 2 (Now re-numbered to Table 1) were the correct values. Values in Lines 160-163 were corrected to reflect the results from the table showing population and individual bias and imprecision.

Reviewer #2comment #12:

Line 182-185: Description the change after re-estimation for Bos et al.’s model as well as both bi-compartmental models from Hodiamont et al. seems to be not accurate: re-estimated volume of distribution was slightly higher than its original value, not remained similar. Similarly, for both bi-compartmental models from Hodiamont et al., re-estimated gentamicin clearance was slightly higher than its original value, not remained similar.

Response to comment #12:

Thank you for your comment. Upon verification, the text was updated in order to reflect the values of PK parameters before and following re-estimation in Table S3.